# Structural Elucidation and Cytotoxicity of a New 17-Membered Ring Lactone from Algerian *Eryngium campestre*

**DOI:** 10.3390/molecules23123250

**Published:** 2018-12-08

**Authors:** Ali Medbouhi, Aura Tintaru, Claire Beaufay, Jean-Valère Naubron, Nassim Djabou, Jean Costa, Joëlle Quetin-Leclercq, Alain Muselli

**Affiliations:** 1Laboratoire de Chimie Organique Substances Naturelles et Analyses (COSNA), Département de Chimie, Faculté des Sciences, Université de Tlemcen, BP 119, Tlemcen 13000, Algeria; sm-ali13@hotmail.fr (A.M.); n_djabou@mail.univ-tlemcen.dz (N.D.); 2Laboratoire Chimie des Produits Naturels (CPN), Campus Grimaldi, Université de Corse, UMR CNRS 6134 SPE, BP 52, 20250 Corte, France; costa@univ-corse.fr; 3Aix Marseille Univ, CNRS, Institut de Chimie Radicalaire, UMR 7273, 13397 Marseille, France; 4UCLouvain, Louvain Drug Research Institute, Pharmacognosy Group, Avenue E. Mounier, 72, bte B1.7203, B-1200 Bruxelles, Belgium; claire.beaufay@uclouvain.be (C.B.); joelle.leclercq@uclouvain.be (J.Q.-L.); 5Centrale Marseille, Aix Marseille Univ, CNRS, FSCM, Spectropole, Marseille 13397, France; jean-valere.naubron@univ-amu.fr

**Keywords:** *Eryngium campestre*, 17-membered ring lactone, cytotoxicity, *Trypanosoma*, *Leishmania*

## Abstract

The chemical composition of a hexanic extract of *Eryngium campestre*, obtained from its aerial parts, was investigated by GC-FID, GC/MS, HRMS, NMR and VCD analyses. The main compounds were germacrene D (23.6%), eudesma-4(15)-7-dien-1-β-ol (8.2%) and falcarindiol (9.4%), which are associated with a new uncommon and naturally found 17-membered ring lactone. This 17-membered ring features conjugated acetylenic bonds, named campestrolide (23.0%). The crude extract showed moderate antitrypanosomal (*Trypanosoma brucei brucei*), antileishmanial (*Leishmania mexicana mexicana*) and anticancer (cancerous macrophage-like murine cells) activities, and also displayed cytotoxicity, (human normal fibroblasts) in similar concentration ranges (IC_50_ = 3.0, 3.9, 4.0 and 4.4 µg/mL respectively). Likewise, campestrolide displayed low activity on all tested cells (IC_50_: 12.5–19.5 µM) except on *Trypanosoma*, on which it was very active and moderately selective (IC_50_ = 2.2 µM. SI= 8.9). In conclusion, the new compound that has been described, displaying a singular structure, possesses interesting antitrypanosomal activity that should be further investigated and improved.

## 1. Introduction

The genus *Eryngium*, the most common of the *Apiaceae* family, comprises more than 250 species with cosmopolitan distribution in temperate regions of all continents, mainly in Eurasia, North Africa and South America [1]. This genus has been the subject of several phytochemical investigations. A remarkable richness in natural chemicals with interesting bioactivities was reported in the literature: terpenoids, polyacetylenes, saponins, steroids and phenolics (such as, flavonoids and coumarins) [2]. This phytochemical diversity can explain the large traditional uses of many *Eryngium* species in the treatment of emetic and gastrointestinal infections [3], several types of inflammatory disorders [4] and various parasitic infections [5]. More traditional uses of many *Eryngium* species include use in antidotes for poisons, hypoglycemic agents [6], diarrhea remedies, stimulants, aphrodisiacs, antitussives and diuretics [7].

Among this large variety of species is the *E. campestre* species (Figure 1), which is a perennial plant that measures from 30 to 60 cm in length. This species is widespread in Western and Central Europe, North Africa, the Middle East and the Caucasus [8]. The plant has been used in European herbal medicine as an infusion for the treatment of whooping cough, as well as in the treatment of kidney and urinary tract inflammations [9].

Many compounds belonging to different phytochemical classes were already identified in this plant. Some of these were flavonoids and flavonoacyl derivatives, extracted from the aerial parts and the roots [10,11,12]; monoterpene glycosides with cyclohexanone moiety [13]; and a coumarin derivative [14] and triterpene saponins were found in the roots [15]. *E. campestre* essential oils displayed a complex chemical composition with hydrocarbon and oxygenated sesquiterpenes. The main components were germacrene D, β-curcumene, (*E*)-β-farnesene, spathulenol, α-bisabolol and α-cadinen-15-al [16,17].

*E. campestre* has been the subject of many biological investigations. The methanol extracts from the aerial parts showed very strong antitumoral activity on potatoes’ tumor cells induced by *Agrobacterium tumefaciens* (ATCC 23341), but no significant antimicrobial activity [18]. The flavonol-rich methanol extracts of *E. campestre* aerial parts exhibited moderate to strong antioxidant activity in DPPH radical scavenging and reducing power assays; in contrast, no effect on Alzheimer’s disease was reported [19]. Ethanol extracts, obtained from the aerial parts and the roots, revealed noticeable anti-inflammatory and antinociceptive activities [20].

Protozoan neglected diseases, such as African trypanosomiasis and cutaneaous leishmaniasis, are tropical infections affecting more than one billion people worldwide and lacking financial investments. The current available treatments suffer from some toxicity, administration difficulty and even resistance development. In this area, the plants and their bioactive secondary metabolites constitute a potential source of crucially needed new and effective drugs [21,22].

As part of our research work on bioactive metabolites from Algerian plants, phytochemical investigation of the hexane extracts obtained from the *E. campestre* aerial parts was performed. This study describes the isolation and structural elucidation of a new 17-membered ring lactone using chromatography techniques (GC-FID, GC-MS), HRMS, 1D and 2D NMR experiments and circular dichroism, along with two known polyacetylenes: falcarinol and falcarindiol. The cytotoxicity and some antiprotozoal activities of the bulk extract and the newly isolated macrocyclic lactone were then investigated in vitro, in order to evaluate their potential pharmacological properties.

## 2. Results and Discussion

### 2.1. Chemical Composition

The analysis of the filtered hexane extract of the *E. campestre* (HEEC) aerial parts using hyphenated methods allowed the identification of 34 components, which accounted for 84.0% of the total composition (Table 1). The chemical composition was dominated by the oxygenated compounds (52.8%). The principal classes were sesquiterpenes (48.6%) and polyacetylenes (35.2%). Hydrocarbon compounds were exclusively represented by sesquiterpenes (31.2%). The main components were germacrene D **9** (23.6%), unknown compounds **33** (23.0%), **34** (9.4%), eudesma-4(15)-7-dien-1-β-ol **27** (8.2%) and falcarinol **32** (2.8%). Thirty-two components were identified by comparison of their RI and MS data with those from our home library, “Arômes”. The spectrometric data of **33** and **34** were not found in our MS-library. Their identifications were achieved after purification by column chromatography, using a combination of analytical techniques: GC-FID, GC/MS-EI, HRMS, exhaustive NMR characterization and VCD. The structures of the main components that were identified are presented in Figure 2. The structure elucidation of **33** will be presented and discussed separately.

### 2.2. Identification of Compounds Not Present in MS-Libraries

Column chromatography was carried out using a gradient of polarity with hexane and diisopropyl ether and then followed by UV detection, producing 75 fractions from the hexane extract of *E. campestre*. Among them, **34** (8 mg of an almost pure sample) was isolated in the fraction **F34** obtained with diisopropyl ether/hexane (10/100). EI mass spectra of **34** exhibited a base peak at *m*/*z* 129 and a signal at *m*/*z* 260, which could be attributed to the molecular ion. ESI (+)-HRMS measurement confirms the molecular formula of C_17_H_24_O_2_ (detected ion C_17_H_24_O_2_Na^+^ (*m*/*z*)*_experimental_* 283.1668 and (*m*/*z*)*_theoretic_* 283.1667; error +0.4 ppm). Additionally, the acquired ^13^C-NMR spectra showed the presence of both oxygenated *sp*^3^ methyne groups detected at 63.50 ppm and 58.61 ppm, respectively, and four *sp* carbon atoms detected between 79.86 ppm and 68.69 ppm. These observations lead us to an oxygenated polyacetylene skeleton, a derivative of falcarinol **32**. A comparison of mass spectra and NMR data in the literature [26] allowed the identification of falcarindiol (Figure 2—**34**). This polyacetylenic compound, accounting for 9.4% of the *E. campestre* hexane extract, is a main component of the roots of *Angelica japonica* [26], *Daucus carota* [27], and *Petroselinum crispum* [Mill.] Nym. ssp. *tuberosum* [25]. Its identification in the *Eryngium* species was first reported by Bohlmann and Zdero [28] in ether extracts of *E. alpinum, E. coeruleum* and *E. giganteum*. However, this is the first time it has been identified in *E. campestre*. Falcarindiol has been shown to have anti-inflammatory, antibacterial and anticancer activities, as well as protective effects against hepatotoxicity [29,30].

Column chromatography using the gradient of solvent diisopropyl ether/hexane (5/95) produces a yellow, odorous, oily compound (**F21**: 13 mg, **33** at 96%). GC/MS-EI spectra of **33** exhibited an isotopic cluster at *m*/*z* 270 and 271, and a base peak at *m*/*z* 115. Other fragment ions were observed at *m*/*z* 55, 91, 128 and 145 (Figure 3). The ESI (+)-HRMS measurement performed on **33** reveals the elemental composition of C_18_H_22_O_2_ (detected ion: C_18_H_22_O_2_Na^+^: (*m*/*z*)_*experimental*_ 293.1513 and (*m*/*z*)_*theoretic*_ 293.1512. error +0.3 ppm), which would correspond to a compound possessing an additional carbon atom and one supplementary unsaturation (DBE: double bounds equivalent = 8) as compared to the falcarindiol **34** (C_17_H_24_O_2_, DBE = 7). The IR spectrum exhibits absorption peaks at 1739 cm^−1^, 1098 cm^−1^ and 1225 cm^−1^, corresponding to C=O and C-O vibrations. This indicates the presence of a cyclic lactone function. Additionally, absorption bands at 2255 cm^−1^ and 2856–2928 cm^−1^ indicated the presence of C≡C and CH_2_ bonds, respectively.

A rapid analysis of the ^1^H- and ^13^C-NMR spectra (Figure 4) showed the absence of -CH_3_ groups and the presence of a ketone group (172.98 ppm). Furthermore, the characteristic resonances of a polyacetylene skeleton (constituted by a sequence of two consecutive triple bonds) are identified using the ^13^C-NMR spectrum in combination with DEPT data (Table 2).

The complete assignment was achieved using the 2D NMR spectra (Appendix A with this article). In addition, NOE correlations showed spatial vicinity among the aliphatic chain and the unsaturated part of the molecule, in particular between 3-CH_2_ → 17-CH and 7-CH_2_ → 10-CH (Appendix A). The combination of all spectroscopic data allowed the recognition of a seventeen-atom macrocyclic lactone and its structure is represented below (Figure 4—inset). This unprecedented 17-membered ring lactone, featuring conjugated acetylenic bonds, has been named **Campestrolide**. Nevertheless, the analytical data do not establish the determination of the absolute configuration of 17-CH, which has since been established by vibrational circular dichroism (VCD).

In the last decade, VCD spectroscopy has become a popular technique for the elucidation of absolute configurations of chiral molecules [31,32,33]. Therefore, in order to determine the stereochemistry of the asymmetrical carbon-17 of Campestrolide, the VCD spectrum of a sample containing 80% of compound **33** was measured in CD_2_Cl_2_. Despite the difficulties encountered during the determination of the absolute configuration of a molecule via the VCD spectrum of a mixture, the majority of the measured bands were in good agreement with those obtained by calculation performed on the (*Z*,*S*)-enantiomer (region I, II and III in the VCD spectrum—Figure 5a). Moreover, for the same spectral regions, a good accordance is also observed between the experimental IR spectrum recorded for the analyzed sample and the theoretical IR spectrum calculated for the (*Z*,*S*)-enantiomer (Figure 5b—region I, II and III). This observation reinforces the assignment of these bands to our molecule. Based on the analysis of the spectral data and taking into account the opposite signs between the measured and calculated VCD spectra, we could conclude that the naturally obtained Campestrolide is detected as (*R*)-enantiomer (Figure 5c), namely (*Z*)-17(*R*)-vinyloxacycloheptadeca-10-en-13,15-diyn-2-one. It should be mentioned that the identified configuration is perfectly consistent with those generally reported in the literature for polyacetylenes derivatives [33,34,35].

To our knowledge, this uncommon natural macrocyclic lactone that includes one cyclo-1,3-diynes motif has only been reported in one other compound (ivorenolide B), which was extracted from *Khaya ivorensis*—a dicotyledone tree from the *Meliaceae* family [33,36]. Furthermore, this compound, as well as its 18-memberded ring lactone analogous ivorenolide A, showed significant immunosuppressive activity [33,37,38,39]. Other macrocyclic lactones (20-membered ring) have been extracted from the marine dinoflagellate *Amphidinium* species. Researchers have reported cytotoxic properties for all the described amphidinolides mentioned above [40].

### 2.3. Biological Activities

Both the crude hexanic extract of *E. campestre* and compound **33** were evaluated for their in vitro cytotoxicity and antiprotozoal activities (Table 3). The plant extract showed similar moderate activities (IC_50_ ≈ 4 µg/mL) on all tested cells and hence, without any selectivity for the concerned parasites. The newly identified macrocyclic lactone was strongly active on *Trypanosoma* (IC_50_ = 0.6 µg/mL or 2.2 µM ≈ 2 µM), as defined by Beaufay et al. (IC_50_ ≤ 2 µM) [41], with a reasonable selectivity (SI = 8.9). As such, it fits the stated criteria for antiparasitic hits [42], i.e., activity in vitro with an IC_50_ < 1 µg/mL and at least ten-fold higher than on a mammalian cell line. As the calculated campestrolide log P value (log P = 5.56) [43] was close to the one reported for falcarinol (log P = 5.50) [44], we can assume good membrane permeability of the macrocyclic lactone. Its newly described structure could also be used as a prototype in structure–activity relation studies to improve activity/selectivity.

The antileishmanial or anticancer activities were notable (IC_50_ = 3.4 and 4.8 µM, respectively) but not as selective as those observed for the extract. Another well-known complex polyene macrolide is amphotericin B (IC_50_ = 0.09 ± 0.06 µM), a reference drug widely used to treat visceral leishmaniasis but still little studied on cutaneaous one and with dose-limiting toxicity [45]. Some patents are also based on different kinds of macrolides as antiparasitics [46].

Campestrolide can partially explain all the activities observed for the crude extract but it is certainly not the only active component of the mixture, as observed with the other samples containing different percentages of campestrolide (data not shown). Therefore, other components, e.g., the identified polyacetylenes, could influence activities by addition or synergism action. This has already been observed, for example, with the polyphenolic compounds [47]. Indeed, natural polyacetylenes are highly bioactive and reactive phytochemicals and were largely isolated from the *Apiaceae* family, mostly due to the aliphatic C_17_ chains. Unstable compounds of this variety are known to exhibit some cytotoxic behaviors and an extract, rich in these unsaturated derivatives, has already been shown to possess synergetic cytotoxic activity in combination with taxol [48]. Previous researchers have also shown such compounds to possess encouraging potential against protozoan diseases. However, this requires further investigation [21]. Falcarindiol, another major compound of the crude extract, has yet to be tested for antitrypanosomal and antileishmanial activity. However, Falcarinol, also called panaxynol, was found to be strongly active and selective (IC_50_ = 0.01 µg/mL, corresponding to 0.04 µM, and SI = 858) on *Trypanosoma brucei brucei* compared to HeLa cytotoxicity. The inhibition of the trypanothione reductase was proposed as one target of this very reactive alkylating agent [49]. Another two natural polyacetylenes, 8-hydroxyheptadeca-1-ene-4,6-diyn-3-yl ethanoate and 16-acetoxy-11-hydroxyoctadeca-17-ene-12,14-diynyl ethanoate, were highly active and selective on *Trypanosoma brucei rhodesiense*, *T. cruzi*, *Plasmodium falciparum* and *Leishmania donovani* axenic and the infected macrophages amastigotes compared on L6 rat skeletal myoblasts (IC_50_ = 0.1–2.5 µM. SI > 10). The terminal methylene double bond seems to improve activity [50].

In addition, germacrene D, another major identified compound, possesses some antiparasitic activity as with other non-lactonized sesquiterpenoids and eudesmanolides, with 61% growth inhibition of the *T. cruzi* trypomastigotes at 100 µg/mL. It also inhibited cruzain, an essential *T. cruzi* cysteine protease, with an IC_50_ of 22.1 µg/mL [21].

Concerning the reported plant extract, dichloromethane and methanol compounds from the aerial parts of the *E. campestre* displayed antileishmanial activity with IC_50_ values of 36 and 15 µg/mL, respectively, on *L. donovani* promastigotes. They were inactive on the *P. falciparum* D6 and W strains [51]. However, the petroleum ether extract exhibited more than 50% *Plasmodium* growth inhibition at 4.81 µg/mL [23]. Other *Eryngium* species have previously shown interesting antiparasitic activities (IC_50_ < 20 µg/mL) on *T. cruzi* [50] and *L. donovani,* but appeared less active or inactive on *Plasmodium falciparum* [52,53]. A daucane sesquiterpene with moderate antileishmanial activity (IC_50_ values of 14.33 and 7.84 µM *on L. tarentolae* promastigotes and *L. donovani* amastigotes respectively) was isolated from *E. foetidum* aerial parts [5].

## 3. Materials and Methods

### 3.1. Plant Material

Aerial parts of *Eryngium campestre* L. (Apiaceae) were collected in June 2016, from the north-western areas of Algeria. The botanical identification of the plants was performed according to the botanical determination keys summarized in the Flora of Algeria [54] and a voucher specimen has been deposited in the COSNA laboratory (Voucher Code: EC1-06-2016).

### 3.2. Extraction

The dried aerial parts (120 g) were powdered and weighed, then extracted with hexane (500 mL) under reflux in a Soxhlet apparatus, at 68 °C. The extract was first filtered on filter paper and then on PTFE filters (0.2 µm). After that, the extract was evaporated to dryness under reduced pressure. This resulted in a yield of 1.05%.

### 3.3. Compound Identification

The identification of individual compounds in the *E. campestre* hexanic extract was carried out using different techniques, such as GC/FID, GC/MS (EI) and 1D and 2D NMR. The identifications were mainly based on comparison of the retention indices (RI) and the MS spectra with those contained in our laboratory library “Arômes”. When spectral data were not present in the “Arômes” library, the RI and the MS data were compared with those from commercial libraries [24,55,56,57]. Compounds absent from consulted libraries were isolated by the fractionation process, using flash column chromatography and then identified mainly by NMR spectroscopy.

### 3.4. Fractionation

We measured 300 mg of the hexanic extract. This was then submitted to chromatography on a silica gel column (200–500 µm, 4 g, Clarisep^®^ Bonna Agela Technologies, Willington, CT, USA), using an Automatized Combi Flash apparatus (Teledyne ISCO. Lincoln., NE, USA) equipped with a fraction collector and monitored by a UV detector. The solvents of elution were *n*-hexane (A) and diisopropyl ether (B), with a gradients of elution: a (A: 100%; B: 0%), b (A: 98%; B: 2%), c (A: 96.5%; B: 3.5%), d (A: 95%; B: 5%), e (A: 90%; B: 10%) and f (A: 0%; B: 100%).

### 3.5. GC-FID Conditions

Analyses were carried out using a Perkin-Elmer Autosystem XL GC apparatus (Walthon, MA, USA), equipped with a dual flame ionization detection (FID) system and fused-silica capillary columns, namely, Rtx-1 (polydimethylsiloxane) and Rtx-wax (polyethyleneglycol) (60 m × 0.22 mm i.d.; of a film thickness of 0.25 µm). The oven temperature was programmed from 60 to 230 °C at 2 °C/min and then held isothermally at 230 °C for 35 min and helium was employed as the carrier gas (1 mL/min). The injector and the detector temperatures were maintained at 280 °C and the samples were injected (0.2 µL of pure oil) in the split mode (1:50). The RI of each compound was determined relative to the retention times of a series of *n*-alkanes (C5–C30) by linear interpolation, using the Van den Dool and Kratz (1963) equation with the aid of the software from Perkin-Elmer (TotalChrom navigator, 6.3.1, Shelton, CT, USA). The relative percentages of the extract constituents were calculated from the GC peak areas by the normalization procedure, without the application of correction factors.

### 3.6. GC/MS-EI Conditions

Samples were analyzed with a Perkin-Elmer Turbo mass detector (quadrupole) coupled to a Perkin-Elmer Autosystem XL, equipped with fused-silica capillary columns Rtx-1 and Rtx-Wax. The oven temperature was programmed from 60 to 230 °C at 2 °C/min and then held isothermally at 230 °C (35 min) and helium was employed as the carrier gas (1 mL/min). The following chromatographic conditions were employed: the injection volume was 0.2 µL of pure oil; the injector injector temperature was 280 °C; split 1:80; the ion source temperature was 150 °C; the ionization energy was 70 eV; the MS (EI) data were acquired over the mass range 35–350 Da; and the scan rate was 1 s.

### 3.7. NMR Conditions

NMR experiments were acquired in CDCl_3_ (EuroIsotop, Saint Aubin, France) at 300 K using a Bruker Avance DRX 500 NMR spectrometer (Karlsruhe, Germany), operating at 500.13 MHz for ^1^H and 125.76 MHz for ^13^C Larmor frequency with a double resonance broadband fluorine observe (BBFO) and 5 mm probehead. ^13^C-NMR experiments were recorded using the one-pulse excitation pulse sequence (90° excitation pulse), with ^1^H decoupling during signal acquisition (performed with WALTZ-16). The relaxation delay was set at 2 s. For each analysed sample, depending on the compound concentration, 3 k up to 5 k free induction decays (FID) 64 k complex data points were collected using a spectral width of 30,000 Hz (240 ppm). Chemical shifts (δ in ppm) were reported relative to the residual signal of CDCl_3_ (δ_C_ = 77.04 ppm and δ_H_ = 7.26 ppm, respectively). Complete ^1^H and ^13^C assignments of the requested compounds were obtained using 2D gradient-selected NMR experiments, ^1^H-^1^H COSY, ^1^H-^13^C HSQC, ^1^H-^13^C HMBC and ^1^H-^1^H NOESY, for which conventional acquisition parameters were used, as described in the literature [58].

### 3.8. High-Resolution Mass Spectrometry Experiments

High-resolution mass spectrometry (HRMS) experiments were performed with a Synapt G2 HDMS quadrupole/time-of-flight (Manchester, UK), equipped with an electrospray source operating in the positive mode, ESI(+). The samples were introduced at a 10 µL min^−1^ flow rate (capillary voltage +2.8 kV, sampling cone voltage +20V under a curtain gas (N_2_), flow of 100 L h^−1^ and heated at 35 °C). The samples were dissolved and further diluted in methanol (Sigma-Aldrich, St.-Louis, MO, USA) doped with sodium chloride (0.1 mM) prior to the analysis. Accurate mass experiments were performed using reference ions from an internal calibration procedure, with two reference ions formed upon electrospray of a poly (ethylene oxide) (PEO). Data analyses were conducted using MassLynx 4.1 programs, which were provided by Waters (Manchester, UK).

### 3.9. VCD Measurements

The infrared (IR) and vibrational circular dichroism (VCD) spectra were recorded on a Bruker PMA 50 accessory, coupled to a Vertex 70 Fourier transform infrared spectrometer (Bruker, Wissembourg, Germany). A photoelastic modulator (Hinds PEM 90, Hids Instruments, Portland, OR, USA) set at l/4 retardation was used to modulate the handedness of the circular polarized light at 50 kHz. Demodulation was performed by a lock-in amplifier (SR830 DSP, Zurich Instruments, Zurich, Switzerland). An optical low-pass filter (<1800 cm^−1^) was used to enhance the signal/noise ratio before use of the photoelastic modulator. A transmission cell of 200 μm optical pathlength, equipped with CaF_2_ windows, was used. The solutions with a concentration of 0.1 mol L^−1^ were prepared by dissolving the solid samples in CD_2_Cl_2_. The VCD spectrum of the enantiomer was measured at room temperature using a sample with an estimated purity of 80%. The baseline of the VCD spectrum was corrected by the subtraction of the VCD spectrum of the solvent. For each individual spectrum, about 16,000 scans were averaged at 4 cm^−1^ resolution (corresponding to 4 h of measurement time). For the IR absorption spectra, the cell filled with CD_2_Cl_2_ served as a reference. The spectra are presented without any smoothing and further data processing.

The calculation of the VCD and IR spectra was performed on the (*S*)-enantiomer of compound **33** using Gaussian 16 package [59]. The geometry optimizations, vibrational frequencies, IR absorption, and VCD intensities were calculated with the Density Functional Theory (DFT) using the B3LYP functional combined with the 6-311G (d,p) basis set. The average solvent (CH_2_Cl_2_) effects have been introduced using the implicit solvation model, SMD, which is based on the integral equation formalism of the polarizable continuum model (ief-pcm). The computed harmonic frequencies are generally larger than the fundamentals observed experimentally. They were calibrated using a scaling factor of 0.98. The IR absorption and the VCD spectra were constructed from the calculated dipole and the rotational strengths assuming the Lorentzian band shape with a half-width at a half-maximum of 8 cm^−1^. The computational method is detailed in the Appendix A with this article.

### 3.10. Biological Assays

The crude extract and the newly identified pure compound (91.9%), campestrolide, were evaluated in vitro for their cytotoxicity on two mammalian cell lines: normal human fibroblasts (WI38) and cancerous macrophage-like murine cells. Their antiprotozoal activities were also investigated on *Trypanosoma brucei brucei* (Tbb. strain 427) bloodstream forms and *Leishmania mexicana mexicana* (Lmm. MHOM/BZ/84/BEL46) promastigotes. The tests were performed using [60,61] stock solutions prepared in DMSO at 10 mg/mL. For the cytotoxicity assay, the samples were tested in eight serial 1.7-fold dilutions (150 µL transferred into 100 µL fresh medium) in the 96-well microtiter plates (concentration range: 0.7–25 µg/mL). The camptothecin (Sigma-Aldrich, St.-Louis, MO, USA) was used as a positive control with 5-fold dilutions (concentration range: 0.00032–25 µg/mL) from a 10 mg/mL DMSO stock solution. For the antiparasitic assays, the samples were tested in eight serial 2-fold dilutions (concentration range: 0.20–25 or 0.10–12 µg/mL on *Leishmania* and *Trypanosoma*, respectively). Suramine sodium and pentamidine isethionate salts (Sigma-Aldrich, St.-Louis, MO, USA) were used as a positive control with 3-fold dilutions (0.0046–10 µg/mL) from a stock solution of 2 mg/mL as well as amphotericin B. A maximum of 0.5% DMSO was previously verified to be non-toxic in all the biological assays. The IC_50_ values were determined using Microsoft Excel and GraphPad Prism 7.0 software (GraphPad, San Diego, CA, USA), based on a nonlinear regression. The selectivity index was calculated in comparison to the WI38 cytotoxicity to assess the therapeutic properties as antiparasitic or anticancer hits.

## 4. Conclusions

A new uncommon macrocyclic lactone named campestrolide, along with other known compounds, was isolated and identified in the *Eryngium campestre* hexane crude extract. This newly described compound presents a particular structure that is composed of a rigid 1,3-dynes motif and a flexible aliphatic part. Moreover, campestrolide possesses relatively strong antitrypanosomal activity but moderate selectivity. Therefore, we assume that campestrolide could be considered for further investigation to improve activity/selectivity. Furthermore, it is partially responsible for the observed antiprotozoal activities and cytotoxicity in the crude extract.

## Figures and Tables

**Figure 1 molecules-23-03250-f001:**
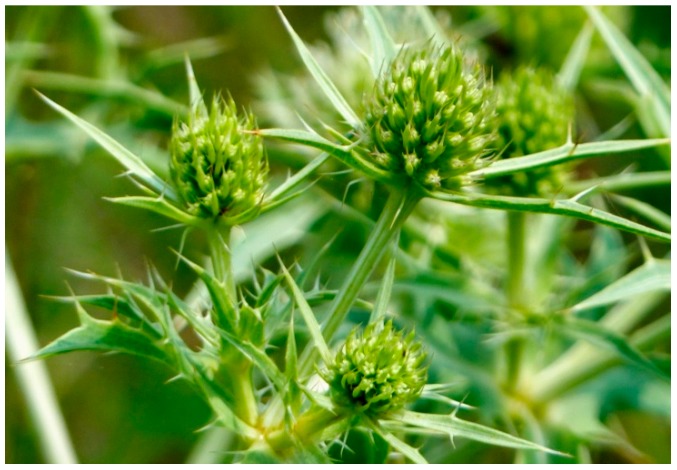
Aerial parts of *Erygium campestre*.

**Figure 2 molecules-23-03250-f002:**
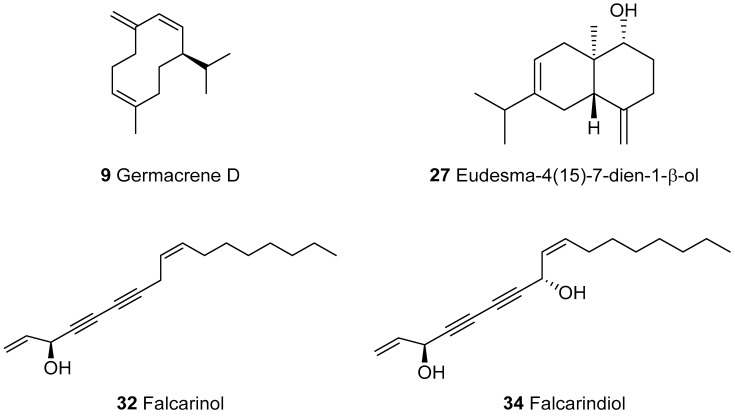
The main components identified in the hexane extract of *E. campestre,* from Algeria.

**Figure 3 molecules-23-03250-f003:**
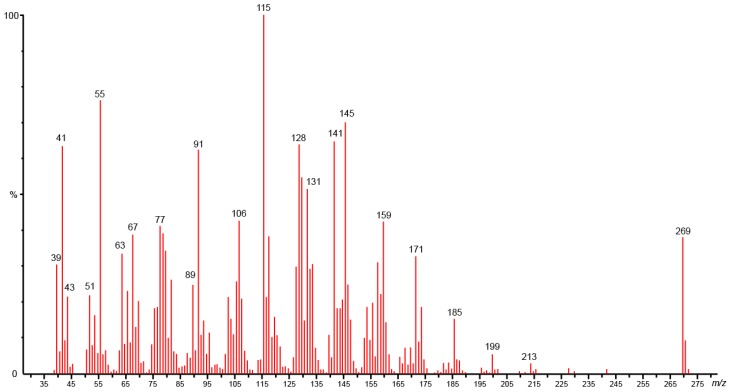
Electronic impact mass spectrum of campestrolide **33** (70 eV).

**Figure 4 molecules-23-03250-f004:**
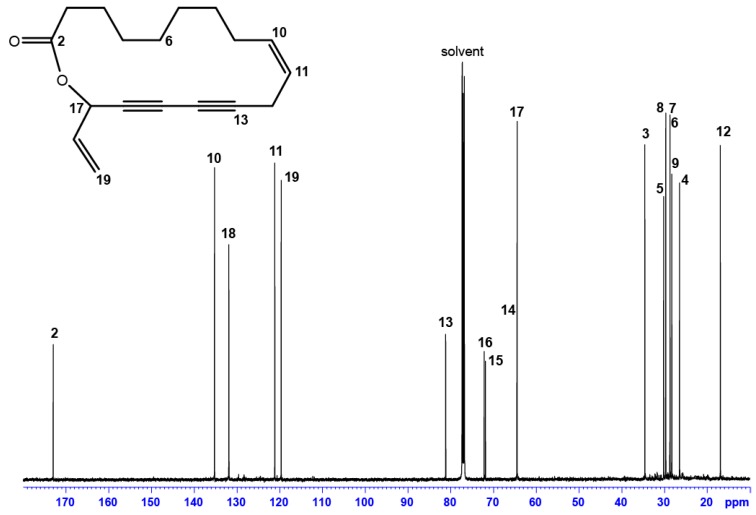
^13^C-NMR spectrum (125.76 MHz in CDCl_3_, 300 K) and structure (inset) of **Campestrolide 33** (N.B. NMR signal assignments refer to the numbering of the molecule illustrated on the inset; the stereochemistry of 17-CH would be revealed subsequent to the VCD experiments).

**Figure 5 molecules-23-03250-f005:**
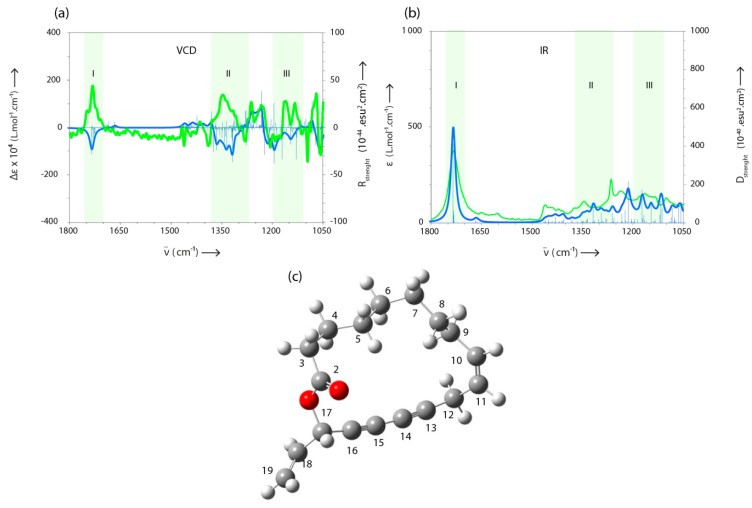
A comparison between the measured (blue) and calculated (green) (**a**) VCD and (**b**) IR spectra, respectively; (**c**) one of the most stable simulated structures of **Campestrolide** (generated for (*Z*,*S*)-enantiomer). N.B. Atoms colors: oxygen in red; carbon in gray; hydrogen in white.

**Table 1 molecules-23-03250-t001:** The chemical composition of the hexane extract of *Eryngium campestre* (HEEC).

No.	Components ^a^	LRIa ^b^	Ria ^c^	RIp ^d^	HEEC ^e^	Identification ^f^
1	α-Copaene	1379	1375	1438	0.4	RI, MS
2	β-Bourbonene	1385	1383	1515	0.1	RI, MS
3	β-Elemene	1388	1387	1589	0.5	RI, MS
4	β-Ylangene	1420	1416	1562	0.6	RI, MS
5	β-Copaene	1431	1432	1581	0.2	RI, MS
6	Alloaromadendrene	1451	1454	1631	tr	RI, MS
7	diepi-4,5-Aristolochene	1467	1465	1665	0.4	RI. MS
8	α-Curcumene	1470	1471	1742	1.0	RI, MS
**9**	**Germacrene D**	**1476**	**1480**	**1704**	**23.6**	RI, MS
10	β-Selinene	1483	1484	1712	0.8	RI, MS
11	α-Muurolene	1496	1503	1720	0.2	RI, MS
12	β-Bisabolene	1500	1500	1720	1.2	RI, MS
13	Sesquicineole	1505	1506	1737	1.1	RI, MS
14	τ-Cadinene	1507	1509	1752	0.2	RI, MS
15	β-Curcumene	1509	1510	1733	0.5	RI, MS
16	δ-Cadinene	1516	1514	1752	1.0	RI, MS
17	α-Cadinene	1535	1533	1743	0.1	RI, MS
18	1,5-Epoxysalvial4(14)-ene	1545	1548	1941	0.8	RI, MS
19	Germacrene B	1553	1551	1827	0.4	RI, MS
20	Spathulenol	1563	1562	2103	1.3	RI, MS
21	β-Copaene-4-α-ol	1575	1573	2141	1.0	RI, MS
22	Salvial-4(14)-en-1-one	1583	1585	2005	0.5	RI, MS
23	Ledol	1600	1602	2030	0.8	RI, MS
24	τ-Cadinol	1632	1638	2169	0.9	RI, MS
25	α-Cadinol	1645	1645	2231	1.6	RI, MS
26	α-Bisabolol	1663	1672	2199	0.7	RI, MS
**27**	**Eudesma-4(15)-7-dien-1-β-ol**	**1681**	**1667**	**2333**	**8.2**	RI, MS
28	14-hydroxy-α-Muurolene	1755	1755	2599	0.3	RI, MS
29	14-hydroxy-τ-Cadinene	1788	1784	2607	0.2	RI, MS
31	Hexadecanoic acid	1942	1941	2930	0.2	RI, MS
32	Falcarinol	2028	2026	-	2.8	RI, MS, [23]
**33**	**Campestrolide**	**-**	**2143**	**2970**	**23.0**	RI, MS, NMR
**34**	**Falcarindiol**	**2190** ^g^	**2164**	**-**	**9.4**	RI, MS, NMR
	Total identification (%)				84.0	
	Hydrocarbon compounds				31.2	
	Oxygenated compounds				52.8	
	Hydrocarbon sesquiterpenes				31.2	
	Oxygenated sesquiterpenes				17.4	
	Non terpenic compounds				3.0	

^a^ Elution order is given on the apolar column (Rtx-1). In bold are the main compounds. ^b^ Retention indices from the literature [24] are on the apolar column, except for **34** [25] (*l*RIa). ^c^ Retention indices are on the apolar Rtx-1 column (RIa). ^d^ Retention indices are on the polar Rtx-Wax column (RIp). ^e^ HEEC: Hexane extract of *Eryngium triquetrum*. The relative percentages of the extract constituents were calculated from the GC peak areas, without application of correction factors. tr = trace (<0.05%). %: Percentages are given on the apolar column, except for components with identical RIa (in such cases, percentages are given on the polar column). ^f^ RI: Retention Indices; MS: Electron Impact Mass Spectrometry; ^g^ The reported value was determined on a different apolar column [25], which would explain the significant difference as compared to our measurement.

**Table 2 molecules-23-03250-t002:** NMR spectroscopic data (500 MHz, CDCl_3_) for **Campestrolide** (**33**).

Position	δ_C_, Type	δ_H_ (J in Hz)	HMBC ^a^
2	172.97, C	-	3, 4, 17, 18
3	34.55, CH_2_	2.41, m	4, 5
		2.38, m	
4	26.45, CH_2_	1.76, m	3, 5, 6
		1.67, m	
5 ^b^	28.68, CH_2_	1.44, m	3, 4, 6, 7
6	29.67, CH_2_	1.28, m	4, 5, 7, 8
7	30.17, CH_2_	1.36, m	5, 8, 9
8 ^b^	28.70, CH_2_	1.44, m	6, 7, 8
9	28.26, CH_2_	2.10, m	8, 10, 11
		2.15, m	
10	135.23, CH	5.62, m	8, 9, 12,
11	121.13, CH	5.52, dt (9.96, 7.53)	9, 12
12	16.91, CH_2_	2.96, dd (18.30; 7.50)	10, 11
		3.07, dd (18.30; 7.50)	
13	81.15, C	-	11, 12
14	64.51, C	-	12, 17
15	71.87, C	-	12, 17
16	72.15, C	-	12, 18, 19
17	64.45, CH	5.92, d (5.85)	18, 19
18	131.88, CH	5.91, ddd (5.85; 10.17; 16.90)	17, 19
19	119.65, CH_2_	5.37, dd (16.90; 1.2)	18
		5.60, dd (10.17; 1.2)	

^a^ HMBC correlations are from proton(s) stated to the indicated carbon; ^b^ the values could be inverted.

**Table 3 molecules-23-03250-t003:** The cytotoxicity (WI38 and J774), antitrypanosomal (Tbb) and antileishmanial (Lmm) activities expressed in the IC_50_ (Mean ± SD in µg/mL and μM for pure compounds from at least six values).

	Cytotoxicity			Antiparasitic Activity				Selectivity Index	
	IC_50_ ± SD in µg/mL (µM for Pure Compound)							IC_50_ WI38/IC_50_ Parasite	
	WI38	J774		Tbb		Lmm		Tbb	Lmm
Hexanic extract	4.44 ± 0.94	4.00 ± 1.07		3.00 ± 0.88		3.86 ± 0.10		1.5	1.2
**33**	5.20 ± 0.24	4.84 ± 0.10		0.59 ± 0.08		3.43 ± 0.02		8.9	1.5
	(19.24 ± 0.87)	(17.89 ± 0.35)		(2.17 ± 0.28)		(12.67 ± 0.09)			
Positive control	0.036 ± 0.022	0.007 ± 0.005		0.031 ± 0.012		0.057 ± 0.008			
	(0.103 ± 0.062) ^a^	(0.021 ± 0.013) ^a^		(0.022 ± 0.008) ^b^		(0.097 ± 0.014) ^c^			

WI38: non cancer human fibroblasts; J774: cancerous macrophage-like murine cells; Tbb*: Trypanosoma brucei brucei* (bloodstream forms); Lmm: *Leishmania mexicana mexicana* promastigotes; Selectivity index calculated for antiparasitic activities compared to WI38 cytotoxicity. Positive control (reference drug): ^a^ camptothecin, ^b^ suramine, ^c^ pentamidine.

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
