# Peer review of "Structural Elucidation and Cytotoxicity of a New 17-Membered Ring Lactone from Algerian Eryngium campestre"

_molecules, 2018, doi:10.3390/molecules23123250_

Round 1

Reviewer 1 Report

This work has analyzed the components of aerial parts of Eryngium campestre extracted by using hexane. A new natural 17-membered ring lactone featuring conjugated acetylenic bonds named campestrolide was isolated and characterized HRMS, NMR and VCD analysis. The antitrypanosomal, antileishmanial and anticancer activities of crude extract and campestrolide had also been investigated. Although the activities of the extract mixture or isolated campestrolide are far less than that of the positive control, this work is certainly meaningful for the finding of new natural compounds and natural product based medicines. I suggest its publication after major revision.

Revision comments:

1.       Page 5, line 129, 131, IR spectra discussion, 1739 nm, 1098 nm and 1225 nm?  It should be cm (-1).

2.       Figure 4, the simulated structure of campestrolide seems not reasonable. Carbon 12-13-14-15-16-17 should be located on the same line (see comments attached chemical 3D structure). VCD and IR have to be recalculated. Also, in the discussion, the calculation method should be mentioned and indicated the calculation details see supporting information. 

Author Response

Response to Reviewer 1 Comments

Point 1: Page 5, line 129, 131, IR spectra discussion, 1739 nm, 1098 nm and 1225 nm?  It should be cm (-1). 

Response 1: The units of IR data have been corrected.

Point 2: Figure 4, the simulated structure of campestrolide seems not reasonable. Carbon 12-13-14-15-16-17 should be located on the same line (see comments attached chemical 3D structure). VCD and IR have to be recalculated. Also, in the discussion, the calculation method should be mentioned and indicated the calculation details see supporting information.

Response 2: We are grateful to review for the provided picture. In contrast, this structure was probably simulated using a molecular mechanic level which does not consider any electronic effects, as in the case of DFT (density functional theory) calculations used for VCD computation purpose. Nevertheless, we redid the geometry optimization calculation using either MP2 level or other functionals. All the results show the torsion of the linear part of the cycle which is due to the combination of two different effects: on one hand, this particular region exhibits a high electron delocalization between the two alkynes that induces a global diminution of the bond lengths; on the other hand, 17-atoms ring exerts on the linear moiety an important tension force contributing to the high deformation observed between atoms 12 to 17. To be mentioned that the recalculated IR and VCD spectra using others theoretical levels are similar to those previously reported and in good agreement with the experimental data.

However, the former Figure 4 (Figure 5 in the new version of the manuscript) has been changed as, by a regrettable mistake, in the previous version of the manuscript, we displayed the data obtained for the (E, S) instead of (Z, S) enantiomer. Anyway, we can note that the isomerisation Z-E does not induce a significant effect on the VCD/IR spectra.

General Comments

All the modifications were underlined in yellow in the text.

Due to the addition of a supplementary figure in the main text of the manuscript, we have reformatted the manuscript layout in order to keep a harmonious pagination. Additionally, the numbering of all the figures has been change, accordingly.

The bibliography was renumbered according to the new reference added to the main text.

Reviewer 2 Report

The manuscript presents a careful chemical analysis of Algerian Eryngium campestre hexanic extract, identification of a new 17-membered ring lactone and determination of its biological activities. Due to discovered antitrypanosomal effects the manuscript is of significant interest to the readers of Molecules. However, the following issues have to be successfully resolved before its publication can be recommended:

1) The Introduction would benefit much by the addition of a photo of Eryngium campestre.

2) The addition or synergism actions mentioned in the subsection 2.3 Biological activities are nicely described in Molecules 2016, 21: 901. Quote and discuss.

3) At what temperature was the Soxhlet extraction performed?

4) Enough details of the Gaussian 16 calculation should be provided in the main text to ensure the reproducibility of the presented results.

5) Typo: Page 5, line 131: Replace indicating by indicated.

Author Response

Response to Reviewer 2 Comments

Point 1: The Introduction would benefit much by the addition of a photo of Eryngium campestre. 

Response 1: A photo of the Eryngium campestre has been included in the introductive part of the manuscript, as suggested by the reviewer 2.  

Point 2: The addition or synergism actions mentioned in the subsection 2.3 Biological activities are nicely described in Molecules 2016, 21: 901. Quote and discuss.

Response 2: The recommended reference has been included and discussed in the section "2.3 Biological activities".  

Point 3: At what temperature was the Soxhlet extraction performed?

Response 3: Temperature of Soxhlet extraction was of 68°C. The detail has also been added to the experimental part section.

Point 4: Enough details of the Gaussian 16 calculation should be provided in the main text to ensure the reproducibility of the presented results.

Response 4: More details of the Gaussian 16 calculation were included in the corresponding experimental section. Nevertheless, the full description of calculation parameters and method could be found in the Supplementary information part.

Point 5: Typo: Page 5, line 131: Replace indicating by indicated.

Response 5: We have made the correction.

General Comments

All the modifications were underlined in yellow in the text.

Due to the addition of a supplementary figure in the main text of the manuscript, we have reformatted the manuscript layout in order to keep a harmonious pagination. Additionally, the numbering of all the figures has been change, accordingly.

The bibliography was renumbered according to the new reference added to the main text.

Round 2

Reviewer 1 Report

The authors have done revisions properly, It is now suitable for publication.

Reviewer 2 Report

The authors successfully addressed all issues raised by this reviewer. Consequently, the manuscript has been significantly improved and can be in its current version recommended for publication in Molecules.